# Learning the delay in delay differential equations

**Robert Stephany & Maria Oprea**
Center for Applied Mathematics
Cornell University
Ithaca, NY 14850, USA
`rrs254@cornell.edu, mao237@cornell.edu`

**Gabriella Torres Nothaft & Mark Walth**
Department of Mathematics
Cornell University
Ithaca, NY 14850, USA
`gt285@cornell.edu, msw283@cornell.edu`

**Arnaldo Rodriguez-Gonzalez**
Department of Mechanical & Aerospace Engineering
University of Buffalo, Buffalo NY, 14260
`arnaldor@buffalo.edu`

**William Clark**
Department of Mathematics
Ohio University
Athens, OH 45701, USA
`clarkw3@ohio.edu`

## Abstract

The intersection of machine learning and dynamical systems has generated considerable interest recently. Neural Ordinary Differential Equations (NODEs) represent a rich overlap between these fields. In this paper, we develop a continuous-time neural network approach based on Delay Differential Equations (DDEs). Our model uses the adjoint sensitivity method to learn the model parameters and delay directly from data. Our approach builds upon recent developments in NODEs and extends earlier neural DDE models, which assume the delay is known a priori. We rigorously justify our adjoint method and use numerical experiments to demonstrate our algorithm's ability to learn delays and parameters from data. Since the delay is rarely known *a. priori*, our approach advances system identification of DDEs from real-world data.

## 1 Introduction and Motivation

Neural ordinary differential equations (NODEs) have proven to be an efficient framework for various problems in machine learning Chen et al. (2018). NODEs assume the map from input to target value can be modeled using a learned ODE. They solve an ODE for the forward pass and use an adjoint approach for backpropagation to update the network parameters. NODEs are a time and memory-efficient model for various regression and classification tasks Dupont et al. (2019). We develop a novel continuous-time machine learning approach based on delay differential equations (DDEs).

DDEs can model a wide range of phenomena: in computer networks, delays arise due to the transmission time of information packets Yu et al. (2004); in gene expression dynamics, a delay occurs due to the time taken for the messenger RNA to copy genetic code and transport macromolecules

from the nucleus to the cytoplasm Busenberg & Mahaffy (1985); in population dynamics, the time taken for a species to reach reproductive maturity introduces a delay Kuang (1993).

There has been recent progress in developing Neural DDEs (NDDEs), the DDE counterpart of NODEs Anumasa & PK (2021), Zhu et al. (2021), Zhu et al. (2022), Zhu et al. (2023). For example, Zhu et al. (2021) and Zhu et al. (2022) show that NDDEs can learn models from time series data that NODEs struggle with. Our work extends Anumasa & PK (2021), Zhu et al. (2021), Zhu et al. (2022), and Zhu et al. (2023), which assume the delay is known *a priori* (they can not learn the delay). By contrast, we derive an adjoint equation that allows our model to learn the delay, parameters, and initial conditions from data. This difference makes our approach more applicable as a system identification tool since the exact value of the delay is often unknown in practice.

In this paper, we propose a novel algorithm to learn the delay and parameters of an unknown DDE using measurements of its solution. Our approach uses the adjoint sensitivity method (see Fig. 1), which we review in section 3.1. We implement our algorithm using `PyTorch` Paszke et al. (2019). We then establish the efficacy of our approach on demonstrative examples in section 4.

## 2 PROBLEM STATEMENT

A DDE with a single delay, $\tau > 0$, can be written in the form:

$$
\begin{aligned}
\dot{x}(t) &= f(x(t), x(t-\tau), \theta) & t \in [0, T] \\
x(t) &= x_0(t) & t \in [-\tau, 0],
\end{aligned}
\tag{1}
$$

where $f : \mathbb{R}^n \times \mathbb{R}^n \times \mathbb{R}^d \to \mathbb{R}^n$. Here, $x_0 : [-\tau, 0] \to \mathbb{R}^n$ is the DDE's initial conditions [*]. Let $C^2([0, T], \mathbb{R}^n)$ denote the set of $C^2$ functions from $[0, T]$ to $\mathbb{R}^n$. Our goal is to learn the parameters $\theta$ and delay $\tau$ from measurements of $x$. We do this using a loss function $J : C^2([0, T], \mathbb{R}^n) \to \mathbb{R}^+$ — with running and terminal cost, $\ell, g : \mathbb{R}^n \to \mathbb{R}^+$, respectively — of the form:

$$
J(x(\cdot)) = \int_0^T \ell(x(t)) \, dt + g(x(T)).
\tag{2}
$$

We consider the following scenario: We have $N_{data}$ measurements, $\{\tilde{x}(t_j)\}_{j=0}^{N_{data}-1}$, of the solution to an unknown DDE of the form of equation 1. Here, $\{t_j\}_{j=0}^{N_{Data}-1}$ are known and partition $[0, T]$ (*i.e.*, $0 = t_0 < t_1 < \cdots < t_{N_{Data-1}} = T$). Finally, we assume the unknown DDE has a constant initial condition given by $\tilde{x}(t_0)$; *i.e.*, $x(t) = \tilde{x}(t_0)$ for $t \in [-\tau, 0]$ [†]. The main task of this paper is to solve the following minimization problem:

**Problem 1** *Find*

$$
\underset{\tau > 0, \; \theta \in \mathbb{R}^d}{\arg\min} \quad J(x(\cdot)) \qquad subject \; to \quad
\begin{cases}
\dot{x}(t) = f\big(x(t), x(t-\tau), \theta\big) & 0 < t \leq T \\
x(t) = x_0, & -\tau \leq t \leq 0.
\end{cases}
$$

## 3 METHODOLOGY

Unfortunately, solving problem 1 exactly is difficult. Therefore, we use an iterative approach based on gradient descent. We accomplish this by employing the adjoint sensitivity method to obtain the gradient of the loss with respect to the model parameters, $\theta$, and the delay, $\tau$.

**Assumption 1** *In this paper, we assume that the running cost $\ell$, the vector field $f$, and the terminal cost $g$ are of class $C^1$. We also assume that the map $(\theta, \tau, t) \to x_{\theta,\tau}(t)$ (where $x_{\theta,\tau}$ is the solution of equation 1 — when the delay is $\tau$ and the parameters are $\theta$ — at time $t$) is of class $C^2$.*

---

[*]Unlike an ODE, where the initial value is a prescribed value, a DDE's initial condition is a function.
[†]This assumption simplifies our analysis but is not essential, see section 5.

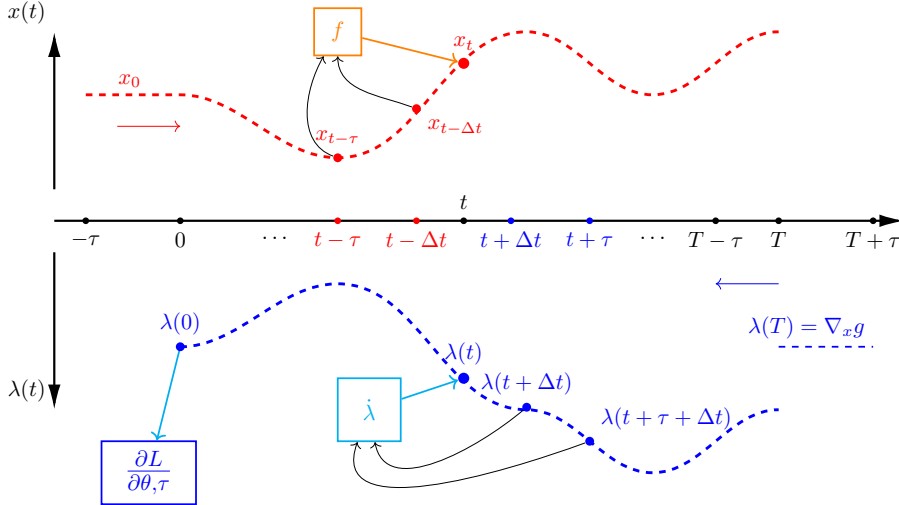

Figure 1: Schematic drawing of the adjoint method for computing gradients. The dashed red curve on the upper half of the figure represents the computed solution to the original delay differential equation $\dot{x}(t) = f(x(t), x(t-\tau), \theta)$. We use the constant initial condition $x(t) = x_0$ for $t \in [-\tau, 0]$. The dashed blue curve on the bottom half represents the computed solution to the adjoint problem.

### 3.1 The adjoint sensitivity method

To derive our result, we use an approach similar to the one for ODEs in Rihan (2021) and Ayed et al. (2019), but tailored for DDEs. We begin by defining the Lagrangian,

$$L(x, \lambda) = J(x) + \int_0^T \langle \lambda(t), \dot{x}(t) - f(x(t), x(t-\tau), \theta) \rangle dt.$$

Here, $\lambda : [0, T] \to \mathbb{R}^n$ is a Lagrange multiplier. We introduce $\lambda$ due to the following observation:

**Remark 1** *Let $x : [-\tau, T] \to \mathbb{R}^n$ be a solution to the DDE, equation 1. Then, for all continuous $\lambda$, $L(x, \lambda) = J(x)$. Hence,*

$$\nabla_\theta J(x) = \nabla_\theta L(x, \lambda) \quad \text{and} \quad \frac{\partial J}{\partial \tau}(x) = \frac{\partial L}{\partial \tau}(x, \lambda).$$

*Since $J$ does not depend on $\lambda$, we can chose a $\lambda$ that makes computing $\nabla_\theta L$ and $\frac{\partial L}{\partial \tau}$ more convenient.*

**Theorem 1** *Let $\dot{x} = f(x(t), x(t-\tau), \theta)$ be a DDE with constant initial condition $x_0(t) = x_0, t \in [-\tau, 0]$ and define the cost as in equation 2. Assume that $f$, $g$, $\ell$, and the flow $x(t)$ satisfy Assumption 1. Choose $\lambda : [0, T] \to \mathbb{R}^n$ to satisfy the following equation:*

$$\begin{cases} \dot{\lambda}(t) &= \nabla_x \ell\left(x(t)\right) - [\partial_x f\left(x(t), x(t-\tau), \theta\right)]^T \lambda(t) \\ &\quad - \mathbb{1}_{t<T-\tau}(t)\left[\partial_y f\left(x(t+\tau), x(t), \theta\right)\right]^T \lambda\left(t+\tau\right) \qquad t \in [0, T] \\ \lambda(T) &= -\nabla_x g\left(x(T)\right) \end{cases} \qquad (3)$$

*Here, $\mathbb{1}_{t<T-\tau}$ is the indicator function on the set $(-\infty, T-\tau)$. Likewise, $\partial_x f$, $\partial_y f$, and $\partial_\theta f$ denote the (Frechet) derivative of $f$ with respect to its first, second, and third arguments, respectively. Then, the derivatives of the cost function with respect to $\theta, \tau$ and $x_0$ are:*

$$\begin{cases} \nabla_\theta J &= -\int_0^T [\partial_\theta f(x(t), x(t-\tau), \theta)]^T \lambda(t) \, dt \\ \frac{\partial J}{\partial \tau} &= \int_0^{T-\tau} \left\langle [\partial_y f(x(t+\tau), x(t), \theta)]^T \lambda(t+\tau), \, \dot{x}(t) \right\rangle \, dt \\ \nabla_{x_0} J &= -\lambda(0) - \int_0^\tau [\partial_y f\left(x(t), x_0, \theta\right)]^T \lambda(t) \, dt \end{cases} \qquad (4)$$

We prove this result in Appendix B.

### 3.2 Proposed Algorithm

We train a model to learn the delay and parameters of the DDE that generated the data. Algorithm 1 summarizes our approach. We begin with random initializations of $\theta$ and $\tau$. Each epoch, we solve

---

**Algorithm 1:** Learning delay and parameters from data

> **Input:** $\{\tilde{x}(t_j)\}_{j=0}^{N_{data}-1}$
> **Output:** $\theta, \tau$

**1 Initialization** *Set $x_0 = \tilde{x}(t_0)$. Select random, initial values for $\theta, \tau$.*

**2 for** $i \in \{1, \ldots, N_{epochs}\}$ **do**

**3**     Solve $\dot{x}(t) = f(x(t), x(t-\tau), \theta)$ using the initial condition $x(t) = x_0$, $t \in [-\tau, 0]$.

**4**     Compute $J(x)$ and check for convergence. Break if converged.

**5**     Otherwise, solve for the adjoint, equation 3

**6**     Compute $\nabla_\theta J(x)$ and $\partial J(x)/\partial \tau$ using equation 4.

**7**     Update $\theta$ and $\tau$ using ADAM.

**8 end for**

---

equation 1 and compute the loss using the model's current $\theta$ and $\tau$ values. We then use this trajectory to solve equation 3 backward in time to find the adjoint. Next, we use equation 4 to compute the gradient of the loss. Finally, we use the Adam optimizer to update $\theta$ and $\tau$ Kingma & Ba (2014). We repeat this for $N_{epochs}$ epochs or until the loss drops below a user-specified threshold.

We implement Algorithm 1 using PyTorch Paszke et al. (2019). We wrap the forward and backward procedures in a torch.autograd.Function class. In the forward pass, we use a second-order Runge–Kutta solver to calculate the predicted trajectory. In the backward pass, we use the same DDE solver to calculate the adjoint by solving equation 3 backward in time. Our code then computes and returns the gradient of the loss with respect to $\tau$ and $\theta$.

In our implementation, we approximate the integral in the loss, equation 2, using a trapezoidal rule. We use SciPy's interpolate function to interpolate between the data points, $\{\tilde{x}(t_j)\}_{j=0}^{N_{data}-1}$. We then evaluate this interpolation and the predicted trajectory at the quadrature points to compute the quadrature rule. Our implementation is open-source and is available at https://github.com/punkduckable/NDDE.

## 4 EXPERIMENTS AND NUMERICAL RESULTS

Here, we test Algorithm 1 by learning the parameters and the delay in a Logistic Delay Equation model. We repeat this analysis on a Delay Exponential Decay model in Appendix A. For these experiments, we use $T = 10$, $g(x) = \|x(T) - \tilde{x}(T)\|_2^2$, and $\ell(x(t)) = \|x(t) - \tilde{x}(t)\|_2^2$, where $\tilde{x}$ denotes the true solution to the hidden DDE (the data set consists of measurements of $\tilde{x}$). Further, we generated all our datasets by solving the corresponding equation using the forward Euler method with a step size of $dt = 0.1$. We use the resulting discretized solution (with 100 data points) as the target dataset, $\{\tilde{x}(t_j)\}_{j=1}^{N_{Data}-1}$, $t_j = j * 0.1$. Finally, we use the Adam optimizer with a learning rate of $0.1$, $\beta_1 = 0.9$, and $\beta_2 = 0.999$ Kingma & Ba (2014).

**Logistic Delay Equation:** The Logistic Delay Equation was first proposed to understand oscillatory phenomena in ecology. It can model the dynamics of a single population growing toward saturation Ruth & Gergely (2020). For this experiment, the true DDE is:

$$\dot{x}(t) = x(t)\left(1 - x(t-1)\right), \qquad\qquad t \in [0, T]$$
$$x(t) = 2 \qquad\qquad t \in [-\tau, 0].$$

We use the model $f\left(x, y, (\theta_1, \theta_2)\right) = \theta_0 x(1 - \theta_1 y)$.

We initialize our model with $\theta_0 = \theta_1 = 1.75$ and $\tau = 1.75$. We then use Algorithm 1 to learn $\theta$ and $\tau$. We use $N_{epochs} = 1,000$ and set the loss threshold at $0.001$. The loss drops below this threshold after just $144$ epochs. Table 1 reports the discovered values of $\theta_0$, $\theta_1$, and $\tau$. Thus, our approach identifies the correct parameters and $\tau$ values. Fig. 2b and 2a depict the loss landscape for this model; they show that the loss has a sink around the true $\theta$ and $\tau$ values. Fig. 2a also exhibits the path our algorithm takes as it trains. Finally, Fig. 3b shows that the target and final predicted trajectories are closely aligned.

Table 1: results for the Logistic Delay Equation with $N_{epochs} = 144$.

|          | True parameters | Discovered parameters |
|----------|-----------------|-----------------------|
| $\theta_0$ | 1             | 1.00325               |
| $\theta_1$ | 1             | 1.00232               |
| $\tau$     | 1             | 0.99347               |

## 5 CONCLUSIONS

In this paper, we presented an algorithm to learn the time delay, parameters, and initial condition from data. Our approach employs the adjoint sensitivity method, suitably adapted for DDEs. We provided a rigorous derivation of our methodology and validated the efficacy of our approach through numerical experiments. In practice, the delay is rarely known a priori. Therefore, our contribution represents a crucial advance in system identification for real-world systems governed by delay differential equations.

There are many potential future research directions to extend our results. We would first like to modify our approach to work with arbitrary initial conditions rather than constant functions. Doing so requires introducing a second Lagrange multiplier, $\mu$, which subsequently changes the adjoint equations, equation 3. Second, from a numerical perspective, we want to understand how robust our algorithm is to limited and noisy data. We would also like to understand how various initialization schemes for $\theta$ and $\tau$ impact this robustness. Lastly, we want to develop theoretical error bounds for the discovered delay and a generalization error for the learned model.

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

## A  FURTHER EXPERIMENTAL RESULTS

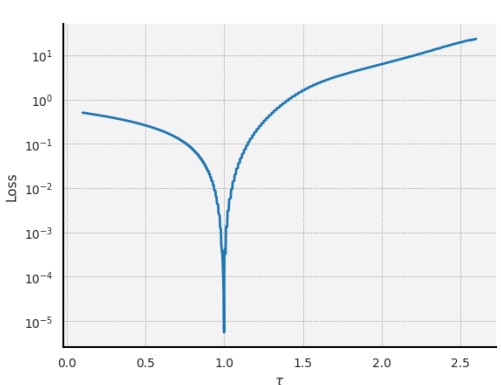
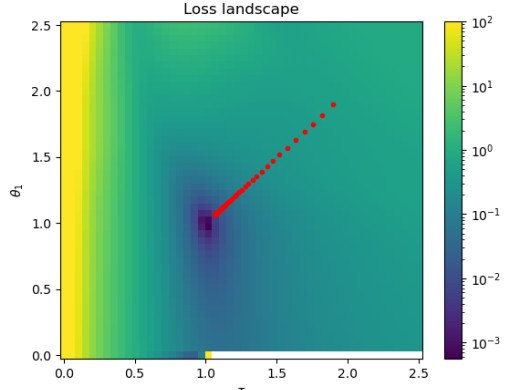

(a) A semi-logarithmic plot of the loss as a function of $\tau$ (with $\theta_1 = \theta_2 = 1$ fixed).

(b) A plot of the log loss for various values of $\theta_1$ and $\tau$ (with $\theta_2 = 1$ fixed). The red dots represent the learned parameters after each epoch of Algorithm 1

Figure 2: Loss landscape for the Logistic Delay Equation experiment.

**Delay Exponential Decay Equation:** The Delay Exponential Decay Equation arises in the study of cell growth, specifically when the cells need to reach maturity before reproducing Rihan & Bocharov (200). For this experiment, the true DDE is:

$$\dot{x}(t) = -2x(t) - 2x(t-1) \qquad\qquad t \in [0, T]$$
$$x(t) = 2 \qquad\qquad t \in [-\tau, 0]$$

For this experiment, we use the model:

$$f(x, y, \theta) = \theta_0 x + \theta_1 y.$$

We initialize our model with $\theta_0 = \theta_1 = -3.0$ and $\tau = 2.0$. We then use Algorithm 1 to learn $\theta$ and $\tau$. Here, we use $N_{epochs} = 1,000$ and a loss threshold of $0.001$. The loss drops below this threshold after 505 epochs. Table 2 reports the discovered values of $\theta_0$, $\theta_1$, and $\tau$. Fig. 3a shows the final predicted and target trajectories. Thus, our approach correctly identifies the model parameters and can faithfully reproduce the target trajectory.

## B  PROOF OF THE ADJOINT EQUATION FOR DDES

In this appendix, we use an adjoint approach to derive an expression for the gradient of $J$ with respect to $\lambda$, $\tau$, and $x_0$.[‡] First, we will consider the gradient with respect to $\theta$. Fix $(\theta, \tau, x_0) \in$

---

[‡]In our experiments, for simplicity, we assume that $x_0$ is known. However, this assumption is not necessary as it is possible to learn this parameter from data. See equation 4.

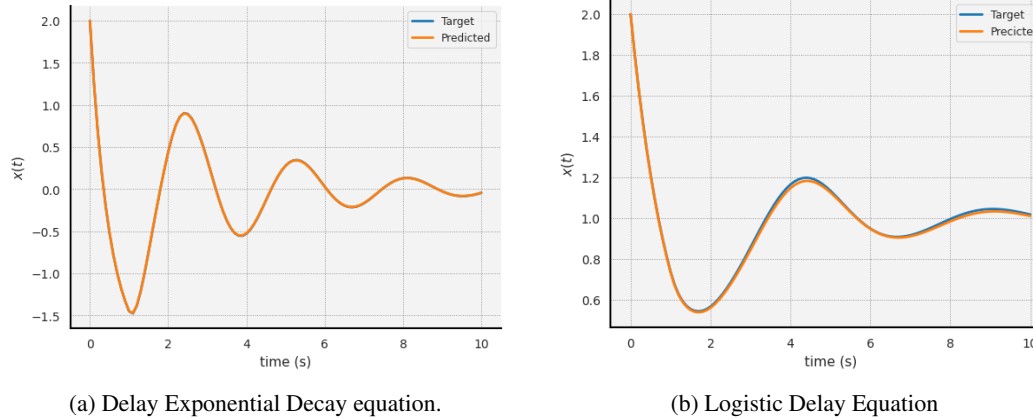

(a) Delay Exponential Decay equation.        (b) Logistic Delay Equation

Figure 3: The final predicted (orange) and target (blue) trajectories in the two experiments.

Table 2: Results for the Delay Exponential Decay Equation with $N_{epochs} = 505$.

|  | True parameters | Discovered parameters |
|---|---|---|
| $\theta_0$ | $-2$ | $-2.02428$ |
| $\theta_1$ | $-2$ | $-2.01691$ |
| $\tau$ | $1$ | $0.99085$ |

$\mathbb{R}^d \times (0, \infty) \times \mathbb{R}^n$. Since $\nabla_\theta J(x) = \sum_{i=1}^{d} (\partial J(x)/\partial \theta_i) \, e_i$, it suffices to consider the case when $\theta \in \mathbb{R}$; we can extend to $\theta \in \mathbb{R}^d$ case by applying $\mathbb{R}$ case component-wise. Throughout this proof, we use results on Frechet differentiation; see Munkres (2018) or an equivalent reference.

Before we proceed, we need some assumptions and notation: To make clear that the solution $x$ depends on $\theta$, we write $x_\theta$ in place of $x$. Further, we assume that $f$, $g$, and $\ell$ are of class $C^1$ (each component function has continuous partial derivatives). Finally, we assume the map $(\theta, t) \to x_\theta(t)$ is of class $C^2$ (each component function has continuous second-order partial derivatives).

Next, let us clarify our derivative notation. This proof involves derivatives of functions of the form $a : \mathbb{R}^n \to \mathbb{R}^m$, $b : \mathbb{R}^n \to \mathbb{R}$, and $c : \mathbb{R} \to \mathbb{R}$. We use the notation $\partial_x a(x) \in \mathbb{R}^{m \times n}$ to denote the matrix of the (Frechet) derivative of $a$ at $x$. If $n = 1$, we treat the $m \times 1$ matrix $\partial_x a(x)$ as a vector in $\mathbb{R}^m$. For the second class, we use the notation $\nabla_x b(y) \in \mathbb{R}^n$ to denote the gradient of $b$ at $y \in \mathbb{R}^n$. Finally, for the third class, we use the notation $\partial c(z)/\partial z \in \mathbb{R}$ to denote the (partial) derivative of $c$ with respect to $z$. Time derivatives are the one exception to this notation: we write $\dot{h}(t)$ for the time derivative $h$, regardless of whether $h$ is real, vector, or matrix-valued.

With that cleared up, let us consider $\partial J(x_\theta)/\partial \theta$. Doing so reveals a problem, which subsequently motivates the Lagrangian. Since the maps $\theta \to x_\theta(T)$ and $u \to g(u)$ are $C^1$, they are differentiable. Therefore, by the chain rule, the map $\theta \to g(x_\theta(T))$ is differentiable with gradient $\langle \nabla_x g(x_\theta(T)), \partial_\theta x_\theta(T) \rangle$. Similarly, for each $t \in [0, T]$, the map $\theta \to \ell(x_\theta(t))$ is differentiable with gradient $\langle \nabla_x \ell(x_\theta(t)), \partial_\theta x_\theta(t) \rangle$. Critically, the components of the latter gradient are continuous on $[0, T]$. Therefore, we are justified in differentiating under the integral sign. In particular, the map $\theta \to \int_0^T \ell(x_\theta(t)) \, dt$ must be differentiable with gradient $\int_0^T \langle \nabla_x \ell(x_\theta(t)), \partial_\theta x_\theta(t) \rangle \, dt$. Therefore, the map $\theta \to J(x_\theta)$ is differentiable with

$$\frac{\partial J}{\partial \theta}(x_\theta) = \langle \nabla_x g(x_\theta(T)), \partial_\theta x_\theta(T) \rangle + \int_0^T \langle \nabla_x \ell(x_\theta(t)), \partial_\theta x_\theta(t) \rangle \, dt.$$

Unfortunately, evaluating this expression requires knowing $\partial_\theta x_\theta(t)$ for each $t \in [0, T]$. Deriving an expression for this quantity is possible but non-trivial. Thankfully, we can address this problem by *adding zero* to $J$ in a clever way: this engenders the Lagrangian

$$L(x_\theta, \lambda_\theta) = J(x_\theta) + \int_0^T \langle \lambda_\theta(t), \dot{x}_\theta(t) - f(x_\theta(t), x_\theta(t - \tau), \theta) \rangle \, dt. \tag{5}$$

Here, $\lambda_\theta : [0, T] \to \mathbb{R}^n$ is a Lagrange multiplier. As we stated in remark 1, $\lambda_\theta$ can be any continuous function. However, we restrict ourselves to $\lambda_\theta$ for which the map $(\theta, t) \to \lambda_\theta(t)$ is of class $C^1$.

We can express the Lagrangian as a single integral using the formula for $J$:

$$L(x_\theta, \lambda_\theta) = \int_0^T \ell(x_\theta(t)) + \langle \lambda_\theta(t), \dot{x}_\theta(t) - f(x_\theta(t), x_\theta(t - \tau), \theta) \rangle \, dt + g(x_\theta(T)) \quad (6)$$

We prove that $L$ is differentiable with respect to $\theta$ and derive a formula for the corresponding gradient. We already know that the maps $\theta \to \int_0^T \ell(x_\theta(t)) \, dt$ and $\theta \to g(x_\theta(T))$ are differentiable. Now, fix $t \in [0, T]$ and consider the map $\theta \to \langle \lambda_\theta(t), \dot{x}_\theta(t) - f(x_\theta(t), x_\theta(t - \tau), \theta) \rangle$. To begin, by assumption, the map $\theta \to x_\theta(t)$ is of class $C^2$. In particular, this means that the component functions of the map $\theta \to (x_\theta(t), x_\theta(t - \tau), \theta) \in \mathbb{R}^n \times \mathbb{R}^n \times \mathbb{R}$ have continuous partials. Thus, this map must be differentiable. Since we assume that $f$ is also of class $C^1$, it must be differentiable. Therefore, by the chain rule, the map $\theta \to f(x_\theta(t), x_\theta(t - \tau), \theta)$ is differentiable with

$$\partial_\theta f(x_\theta(t), x_\theta(t - \tau), \theta) = (\partial_x f(t)) \partial_\theta x_\theta(t) + (\partial_y f(t)) \partial_\theta x_\theta(t - \tau) + \partial_\theta f(t).$$

For brevity, let $f(t)$ be an abbreviation of $f(x_\theta(t), x_\theta(t - \tau), \theta)$. Since $x_\theta$ is of class $C^2$, the map $\theta \to \dot{x}_\theta(t)$ must be differentiable as well. Further, since $(t, \theta) \to x_\theta(t)$ is $C^2$, the equality of mixed partials tells us that

$$\partial_\theta \dot{x}_\theta(t) = \dot{\overline{\partial_\theta x_\theta}}(t).$$

Finally, since $\lambda_\theta$ is of class $C^1$, the component functions of both arguments of the inner product $\langle \lambda_\theta(t), \dot{x}_\theta(t) - f(x_\theta(t), x_\theta(t - \tau), \theta) \rangle$ have continuous partial derivatives with respect to $\theta$. Therefore, the map $\theta \to \langle \lambda_\theta(t), \dot{x}_\theta(t) - f(x_\theta(t), x_\theta(t - \tau), \theta) \rangle$ must be differentiable with

$$\frac{\partial}{\partial \theta} \langle \lambda_\theta(t), \dot{x}_\theta(t) - f(t) \rangle =$$
$$\langle \partial_\theta \lambda_\theta(t), \dot{x}_\theta(t) - f(x_\theta(t), x_\theta(t - \tau), \theta) \rangle$$
$$+ \left\langle \lambda_\theta(t), \dot{\overline{\partial_\theta x_\theta}}(t) - (\partial_x f(t)) \partial_\theta x_\theta(t) - (\partial_y f(t)) \partial_\theta x_\theta(t - \tau) - \partial_\theta f(t) \right\rangle.$$

Crucially, since $\dot{x}_\theta(t) = f(x_\theta(t), x_\theta(t - \tau), \theta)$, the first term in this expression vanishes. Thus,

$$\frac{\partial}{\partial \theta} \langle \lambda_\theta(t), \dot{x}_\theta(t) - f(t) \rangle =$$
$$\left\langle \lambda_\theta(t), \dot{\overline{\partial_\theta x_\theta}}(t) - (\partial_x f(t)) \partial_\theta x_\theta(t) - (\partial_y f(t)) \partial_\theta x_\theta(t - \tau) - \partial_\theta f(t) \right\rangle. \quad (7)$$

By inspection, each component of each argument of the inner product on the right side of equation 7 is continuous. Therefore, we are justified in differentiating under the integral. In particular, the map $\theta \to L(x_\theta, \lambda_\theta)$ must be differentiable with gradient

$$\frac{\partial}{\partial \theta} L(x_\theta, \lambda_\theta) = \int_0^T \langle \nabla_x \ell(x_\theta(t)), \partial_\theta x_\theta(t) \rangle + \left\langle \lambda_\theta(t), \dot{\overline{\partial_\theta x_\theta}}(t) \right\rangle$$
$$- \langle \lambda_\theta(t), (\partial_x f(t)) \partial_\theta x_\theta(t) + (\partial_y f(t)) \partial_\theta x_\theta(t - \tau) + \partial_\theta f(t) \rangle \, dt \quad (8)$$
$$+ \langle \nabla_x g(x_\theta(T)), \partial_\theta x_\theta(T) \rangle.$$

To proceed, we need to make a few simplifications to the expression above. We will focus on two parts of the integrand, which we colored teal and violet. We begin with the teal portion of the integrand. Using integration by parts,

$$\int_0^T \left\langle \lambda_\theta(t), \dot{\overline{\partial_\theta x_\theta}}(t) \right\rangle \, dt = \langle \lambda_\theta(T), \partial_\theta x_\theta(T) \rangle - \langle \lambda_\theta(0), \partial_\theta x_\theta(0) \rangle - \int_0^T \left\langle \dot{\lambda}_\theta(t), \partial_\theta x_\theta(t) \right\rangle \, dt$$
$$= \langle \lambda_\theta(T), \partial_\theta x_\theta(T) \rangle - \int_0^T \left\langle \dot{\lambda}_\theta(t), \partial_\theta x_\theta(t) \right\rangle \, dt. \quad (9)$$

To get the second equality, note that $x_\theta(0) = x_0$. Since $x_0$ is a constant, $\partial_\theta x_\theta(0) = 0 \in \mathbb{R}^n$. We can now move to the violet portion of the integrand. In particular,

$$
\int_0^T \langle \lambda_\theta(t), (\partial_y f(t)) \, \partial_\theta x_\theta(t - \tau) \rangle \ dt
$$

$$
= \int_0^\tau \langle \lambda_\theta(t), (\partial_y f(t)) \, \partial_\theta x_\theta(t - \tau) \rangle \ dt + \int_\tau^T \langle \lambda_\theta(t), (\partial_y f(t)) \, \partial_\theta x_\theta(t - \tau) \rangle \ dt
$$

$$
= \int_0^{T-\tau} \langle \lambda_\theta(t + \tau), (\partial_y f(t + \tau)) \, \partial_\theta x_\theta(t) \rangle \ dt
$$

$$
= \int_0^T \mathbb{1}_{t < T - \tau}(t) \langle \lambda_\theta(t + \tau), (\partial_y f(t + \tau)) \, \partial_\theta x_\theta(t) \rangle \ dt. \tag{10}
$$

To get from the second to the third line, we observe that for $t < \tau$, $x_\theta(t - \tau) = x_0$, which means that $\partial_\theta x_\theta(t) = 0$. Thus, the blue integral is zero. In the red integral, we redefine $t$ as $t - \tau$. In equation 10, $\mathbb{1}_{t < T-\tau}(t)$ is the indicator function of the set $(-\infty, T - \tau]$. Substituting equation 9 and equation 10 into equation 8 gives

$$
\frac{\partial}{\partial \theta} L(x_\theta, \lambda_\theta) = \int_0^T \langle \nabla_x \ell(x_\theta(t)), \partial_\theta x_\theta(t) \rangle - \left\langle \dot{\lambda}_\theta(t), \partial_\theta x_\theta(t) \right\rangle
$$

$$
- \langle \lambda_\theta(t), (\partial_x f(t)) \, \partial_\theta x_\theta(t) \rangle + \mathbb{1}_{t < T - \tau}(t) \langle \lambda_\theta(t + \tau), (\partial_y f(t + \tau)) \, \partial_\theta x_\theta(t) \rangle
$$

$$
+ \langle \lambda_\theta(t), \partial_\theta f(t) \rangle \ dt
$$

$$
+ \langle \nabla_x g(x_\theta(T)) + \lambda_\theta(T), \partial_\theta x_\theta(T) \rangle
$$

$$
= \int_0^T \Big\langle \nabla_x \ell(x_\theta(t)) - \dot{\lambda}_\theta(t) - [\partial_x f(t)]^T \lambda_\theta(t) -
$$

$$
\mathbb{1}_{t < T - \tau}(t) \, [f_y(t + \tau)]^T \lambda_\theta(t + \tau), \partial_\theta x_\theta(t) \Big\rangle \, dt \tag{11}
$$

$$
+ \langle \nabla_x g(x_\theta(T)) + \lambda_\theta(T), \partial_\theta x_\theta(T) \rangle + \int_0^T \langle \lambda_\theta(t), \partial_\theta f(t) \rangle \ dt.
$$

Thus far, our only assumption about $\lambda_\theta$ is that the map $(t, \theta) \to \lambda_\theta(t)$ is $C^1$. Hence, we are free to choose any $\lambda_\theta$. In particular, we can select $\lambda_\theta$ to satisfy the following differential equation:

$$
\begin{cases}
\dot{\lambda}_\theta(t) = & \nabla_x \ell(x_\theta(t)) - [\partial_x f(x_\theta(t), x_\theta(t - \tau), \theta)]^T \lambda_\theta(t) \\
& - \mathbb{1}_{t < T - \tau} [\partial_y f(x_\theta(t + \tau), x_\theta(t), \theta)]^T \lambda_\theta(t + \tau) \\
\lambda(T) = & -\nabla_x g(x_\theta(T))
\end{cases} \tag{12}
$$

We refer to this choice of $\lambda_\theta$ as the adjoint. Substituting the adjoint into equation 11 gives

$$
\frac{\partial}{\partial \theta} J(x_\theta) = \frac{\partial}{\partial \theta} L(x_\theta, \lambda_\theta) = \int_0^T \langle \lambda_\theta(t), \partial_\theta f(x_\theta(t), x_\theta(t - \tau), \theta) \rangle \ dt. \tag{13}
$$

To generalize this to a vector valued $\theta$, notice that if we replace $\theta$ with $\theta_i$ in equation 13, the integral on the right is the i'th component of $\int_0^T [\partial_\theta f(x_\theta(t), x_\theta(t - \tau), \theta)]^T \lambda_\theta(t) \, dt$. Thus,

$$
\nabla_\theta J(x_\theta(t)) = \int_0^T [\partial_\theta f(x_\theta(t), x_\theta(t - \tau), \theta))]^T \lambda_\theta(t) \, dt. \tag{14}
$$

With this established, let us now turn our attention to $\tau$. As we did with $\theta$, we will fix $(\theta, \tau, x_0) \in \mathbb{R}^d \times (0, \infty) \times \mathbb{R}^n$. We will also denote the solution, $x$, by $x_\tau$ to make its dependence on $\tau$ explicit. Using the same argument we used in the $\theta$ case, we introduce a Lagrange multiplier $\lambda_\tau : [0, T] \to \mathbb{R}^n$ with the restriction that the map $(\tau, t) \to \lambda_\tau(t)$ is of class $C^1$:

$$
L(x_\tau, \lambda_\tau) = J(x_\tau) + \int_0^T \langle \lambda_\tau(t), \dot{x}_\tau(t) - f(x_\tau(t), x_\tau(t - \tau), \theta) \rangle \ dt \tag{15}
$$

We now argue that $L$ is differentiable with respect to $\tau$. We can reuse most of our argument from the $\theta$ case with one caveat: we need to re-established the differentiability of the map

$\tau \to f(x_\tau(t), x_\tau(t-\tau), \theta)$ (for a fixed $t$). Before, we proved this by showing that each of $f$'s arguments is differentiable. By assumption, the maps $\tau \to x_\tau(t)$ and $\tau \to \theta$ (the latter being a constant) are differentiable. However, the second argument, $x_\tau(t-\tau)$, depends on $\tau$ both explicitly (in its argument, $t-\tau$) and implicitly (because changing $\tau$ changes the DDE solution). However, this map is the composition of the map $\tau \to (\tau, t-\tau)$ with the map that sends $(\tau, t)$ to the solution of the DDE at time $t$ when the delay is $\tau$. By assumption, both maps are $C^1$ and thus differentiable. Hence, by the chain rule, the map $\tau \to x_\tau(t-\tau)$ is differentiable with derivative $\partial_\tau x_\tau(t-\tau) - \dot{x}_\tau(t-\tau)$. Having established this map is differentiable, we can conclude (again by the chain rule) that the map $\tau \to f(x_\tau(t), x_\tau(t-\tau), \theta)$ is differentiable. We can now proceed exactly as in the $\theta$ case to conclude that $L$ is a differentiable function of $\tau$. In particular,

$$\frac{\partial L}{\partial \tau}(x_\tau, \lambda_\tau) = \int_0^T \langle \nabla_x \ell(x_\tau(t)), \partial_\tau x_\tau(t) \rangle + \left\langle \lambda_\tau(t), \partial_\tau \dot{x}_\tau(t) \right\rangle$$
$$- \langle \lambda_\tau(t), (\partial_x f(t)) \, \partial_\tau x_\tau(t) + (\partial_y f(t)) \, \partial_\tau x_\tau(t-\tau) - (\partial_y f(t)) \, \dot{x}_\tau(t-\tau) \rangle \; dt$$
$$\tag{16}$$
$$+ \left\langle \nabla_x g\left(x_\tau(T)\right), \partial_\tau x_\tau(T) \right\rangle.$$

We can treat the teal and violet portions of the integrand as we did before. Further, since $x_\tau(t-\tau) = x_0$ for $t < \tau$, the orange portion of the integrand is zero in $[0, \tau]$. Therefore,

$$\frac{\partial L}{\partial \tau}(x_\tau, \lambda_\tau) = \int_0^T \left\langle \nabla_x \ell\left(x_\tau(t)\right) - \dot{\lambda}_\tau(t) - [\partial_x f(t)]^T \lambda_\tau(t) - \right.$$
$$\left. \mathbb{1}_{t < T-\tau}(t) \, [f_y(t+\tau)]^T \lambda_\tau(t+\tau), \partial_\tau x_\tau(t) \right\rangle \, dt \tag{17}$$
$$+ \left\langle \nabla_x g\left(x_\tau(T)\right) + \lambda_\tau(T), \partial_\tau x_\tau(T) \right\rangle + \int_\tau^T \left\langle \lambda_\tau(t), (\partial_y f(t)) \, \dot{x}_\tau(t-\tau) \right\rangle \, dt.$$

Thus, if we select $\lambda_\tau$ to satisfy the same adjoint equations as before, equation 12, then we conclude (after a change of variables in the orange integral) that

$$\frac{\partial J(x_\tau)}{\partial \tau} = \frac{\partial L}{\partial \tau}(x_\tau, \lambda_\tau) = \int_0^{T-\tau} \left\langle [\partial_y f\left(x_\tau(t+\tau), x_\tau(t), \theta\right)]^T \lambda_\tau(t+\tau), \dot{x}_\tau(t) \right\rangle \, dt \tag{18}$$

We can use a nearly identical argument to prove that $J$ is a differentiable function of $x_0$. Specifically,

$$\nabla_{x_0} J(x_{x_0}) = -\lambda_{x_0}(0) - \int_0^\tau [f\left(x_{x_0}(t), x_0, \theta\right)]^T \lambda_{x_0}(t) \, dt. \tag{19}$$

Here, $\lambda_{x_0}$ also satisfies equation 12. Dropping the $\theta$, $\tau$, and $x_0$ subscripts in equation 14, equation 18, and equation 19, respectively, gives equation 4. $\square$

