# OpenReview forum: "Learning The Delay in Delay Differential Equations"
_ICLR.cc/2024/Workshop/AI4DiffEqtnsInSci — AI4DiffEqtnsInSci @ ICLR 2024 Poster_

### Official Review · Reviewer_67cL · 2024-02-23
**Important contribution to extend neural ODEs to delay differential equations**

**Rating:** 8
**Confidence:** 3

**Review:**

The authors show a new method learn the delay parameter alongside 'normal' parameters of neural delay differential equations by proposing  a new algorithm to calculate the adjoint for neural delay differential equations. With two simple examples, the authors prove that their algorithm can retrieve the data generating delay parameter and normal parameters.

This work could be a great starting point for developing neural delay differential equations further. The authors give some examples for systems governed by delay differential equations and I agree that the topic is highly relevant. Here, the relevance could have highlighted better by finding prominent examples from different domains. The work is original and can advance the use of neural differential equations to new areas by allowing for delay parameters. A point that may be missing is an ablation experiment that checks if a delay of 0 is found when the neural delay differential equations is confronted with data generated with an ODE.

Pro:
- extends neural ODEs to delay differential equations
- seems to be the first time that adjoint for neural delay differential equations is described
- delay parameter can for the first time be estimated in neural delay differential equations

Con:
- some sloppiness with respect to formatting citations - (Author, Year) formatting is not used when appropriate. This makes reading a bit tedious at times.

---

### Official Review · Reviewer_nj7p · 2024-02-27
**Review of "Learning The Delay in Delay Differential Equations"**

**Rating:** 7
**Confidence:** 2

**Review:**

# Summary

The paper derives an adjoint method that allows the delay in neural delay differential equations to be learned rather than specified *a priori*.
In experiments on simple systems, the proposed algorithm is able to recover the correct parameters and delay from the training data.

# Strengths
1. (**significance**)
The paper builds well upon previous work and addresses an interesting, impactful problem.
2. (**quality**)
The main contribution of the paper is the proof of the adjoint equation in Appendix B, which is thorough and well written, although I haven't verified its correctness.
3. (**clarity**)
More generally, the presentation of the paper, including figures, theorems, and algorithms, is excellent.
4. (**completeness**)
The authors set out extensive and clear opportunities for further work.

# Weaknesses
1. (**completeness**)
While the question of "limited and noisy data" is mentioned briefly in the conclusion, I believe this is a fairly fundamental issue for the method that deserves detailed discussion.
In particular, the method requires interpolating between data points, since the delay $\tau$ is treated as a continuous variable while the data is necessarily discrete.
I strongly suspect that the interpolation may struggle in the presence of noisy, sparse, or irregularly sampled data.
Therefore, a complete discussion of the limitations of this method should include a detailed examination of this point.
1. (**completeness**)
The experiments, while a reasonable proof of concept, are perhaps somewhat trivial, consisting each of only two learnable parameters in addition to the delay.
I would be very interested to know how well the proposed algorithm performs in the presence of many more parameters; for example, in [1] the authors train NDDEs with $\mathcal{O}(100k)$ parameters.
1. (**completeness**)
Another discussion missing from the paper is the possibility of using a *discretize-then-optimize* approach to training NDDEs, i.e., applying automatic differentiation directly to the operations of the numerical ODE solver, as opposed to *optimize-then-discretize*, i.e., adjoint sensitivity analysis; see [2].
While I'm not certain of this, it may be possible to skip step 5 of Algorithm 1 entirely and do step 6 using automatic differentiation instead of using Eq. 4.
At the very least, I think it would be interesting to compare such an approach to the adjoint method derived in this paper.
1. (**completeness**)
There appears to be some closely related work that is not mentioned in the paper [3].

# Other Comments
1. A link to a GitHub repository is included on page 3, but I didn't click on it and I don't think any identifying information about the authors can be determined from the URL.
1. One easy-to-fix problem with the paper's presentation is the bibliography, where some references are given to arXiv preprints instead of the relevant conference proceedings.

# Conclusion
Overall, I like this paper.
The weaknesses discussed above are intended only as constructive criticism that might benefit future versions of this work.

# Citations
[1] Qunxi Zhu, Yao Guo, and Wei Lin. Neural Delay Differential Equations. In *International Conference on Learning Representations*, 2021.

[2] Patrick Kidger. On neural differential equations. arXiv preprint arXiv:2202.02435, 2022.

[3] Xunbi A. Ji, Gábor Orosz. Learning Time Delay Systems with Neural Ordinary Differential Equations. IFAC-PapersOnLine, Volume 55, Issue 36, 2022.

---

### Meta-Review · Area_Chair_9Mrd · 2024-02-28

**Recommendation:** Accept (Poster)

**Metareview:**

Dear Authors,

Thank you for submitting the draft.

Both reviewers agree that the presented work presents interesting strengths. It is expected that authors will be addressing comments by the reviewers in the final draft.

regards

AC

---

### Decision · Program_Chairs · 2024-02-29

Accept (Poster)